# INO80 regulates chromatin accessibility to facilitate suppression of sex-linked gene expression during mouse spermatogenesis

Prabuddha Chakraborty[1], Terry Magnuson[1,2]*

1 Department of Genetics, University of North Carolina at Chapel Hill, Chapel Hill, North Carolina, United States of America, 2 Lineberger Comprehensive Cancer Center, University of North Carolina at Chapel Hill, Chapel Hill, North Carolina, United States of America

* tmagnuson@unc.edu

**Data Availability Statement:** The data supporting this study's findings are publicly available from the GEO database (GSE179584, GSE221682, and GSE273472).

## Abstract

The INO80 protein is the main catalytic subunit of the INO80-chromatin remodeling complex, which is critical for DNA repair and transcription regulation in murine spermatocytes. In this study, we explored the role of INO80 in silencing genes on meiotic sex chromosomes in male mice. INO80 immunolocalization at the XY body in pachytene spermatocytes suggested a role for INO80 in the meiotic sex body. Subsequent deletion of *Ino80* resulted in high expression of sex-linked genes. Furthermore, the active form of RNA polymerase II at the sex chromosomes of *Ino80*-null pachytene spermatocytes indicates incomplete inactivation of sex-linked genes. A reduction in the recruitment of initiators of meiotic sex chromosome inhibition (MSCI) argues for INO80-facilitated recruitment of DNA repair factors required for silencing sex-linked genes. This role of INO80 is independent of a common INO80 target, H2A.Z. Instead, in the absence of INO80, a reduction in chromatin accessibility at DNA repair sites occurs on the sex chromosomes. These data suggest a role for INO80 in DNA repair factor localization, thereby facilitating the silencing of sex-linked genes during the onset of pachynema.

## Author summary

Chromatin accessibility is required for many DNA-protein interactions. Chromatin remodelers are the group of protein complexes that ensure the localized and timely unpacking of DNA to ensure access of a protein to its target DNA regions. The chromatin remodeler, INO80, has been implicated in many cellular processes, including transcription and DNA repair. We report that INO80 regulates the suppression of gene expression at the sex chromosomes in meiotic germ cells during mammalian spermatogenesis. INO80 localizes at the sex chromosomes during the pachynema stage, and its absence leads to a lack of suppression of gene expression at the sex chromosomes, which is known to be detrimental to meiotic progression. These genes were actively transcribed during pachynema in *Ino80*-mutant spermatocytes. DNA damage repair factors such as ATR and MDC1, which are instrumental in suppressing gene expression at the sex chromosomes

**Funding:** T.M. received a grant (R01GM101974) from the National Institute of General Medicine (https://www.nigms.nih.gov) for this study, including salary support for T.M. and P.C. The funders had no role in study design, data collection and analysis, publication decisions, or manuscript preparation.

**Competing interests:** The authors have declared that no competing interests exist.

during pachynema, were not localized properly in *Ino80*-mutant spermatocytes. Chromatin accessibility was also reduced at the DNA damage sites in sex chromosomes of the *Ino80* mutants, which suggests a role for INO80 in the appropriate localization of DNA damage factors at accessible target sites to aid in the silencing of sex-linked genes during pachynema.

## Introduction

Mammalian gametogenesis involves the meiotic division of diploid germ cells that undergo homologous chromosome synapsis and recombination to generate haploid gametes. Extended prophase I during meiosis ensures the faithful execution of recombination to shuffle genetic material between homologous chromosomes by controlled introduction of DNA double-strand breaks (DSB) at the DSB hotspots by SPO11 followed by their repair [1]. Synapsis between homologous autosomes occurs during the zygotene stage of meiotic prophase I. In contrast, non-homologous regions of the X and Y sex chromosomes in male germ cells do not synapse with each other [2].

During pachynema, DNA double-strand break repair (DSBR) factors no longer localize to autosomes, indicating the completion of DSBR. For the sex chromosomes, the DSBR factors sequester at unpaired regions, forming a unique chromatin domain called the sex body [3,4]. The sex chromosomes in spermatocytes undergo modifications by several DNA damage repair (DDR) factors such as BRCA1, ATR, and its activator TOPBP1 at the unpaired (asynapsed) chromosome axes [5,6]. The sex chromosomes also undergo chromatin remodeling to induce epigenetic silencing of sex-linked genes. This process is known as meiotic sex chromosome inactivation (MSCI) [7]. Incomplete MSCI leads to germ cell death during pachynema, causing male infertility [7].

MSCI is initiated by DNA double-strand breaks (DSB) at the asynapsed axes and the incorporation of DSBR proteins at these DSB sites [8,9]. ATR localizes to the DSBs along the chromosome axes [10]. Serine-139 phosphorylation of H2A.X (γH2A.X) at the DSBs by ATR leads to the recruitment of another DSB factor, MDC1 [6,10–12]. MDC1 localization amplifies γH2A.X in the protruding chromatin loops along the axes, establishing the characteristic sex-body and MSCI [6,13].

Chromatin remodelers regulate chromatin accessibility. They play roles in several cellular processes, including DNA repair and transcription regulation. Chromatin remodeling enzymes can hydrolyze ATP to change chromatin conformation by moving, evicting, or incorporating nucleosomes. They can also facilitate specific histone variant exchange to change local chromatin accessibility. There are four major families of chromatin remodeling complexes, all of which play critical functions in murine spermatogenesis [14–20]. INO80 is the main ATPase subunit of the INO80 complex, which has broad effects on several cellular processes. INO80 is important in various DNA metabolic processes, including DNA replication, transcription, repair, and genome stability in several organisms [21]. The exchange and turnover of histone variant H2A.Z is regulated by INO80 [22–25]. INO80 facilitates the recruitment of RNA polymerase II (RNAPII) to the promoters of pluripotency network genes in ES cells [26,27]. In yeast, INO80 is also implicated in removing and degrading ubiquitinated RNAPII from chromatin [28]. Like other chromatin remodelers, INO80 is expressed in several mammalian tissues, including the testis [18], and was reported to facilitate development in a context-dependent manner [18,27,29,30]. However, it is unclear whether the chromatin remodeler INO80 plays any role in the meiotic silencing of the sex chromosomes.

Here, we explore the role of INO80 in silencing meiotic sex chromosomes. We show that INO80 interacts with and regulates sex-linked gene silencing in pachytene spermatocytes. Further, INO80 promotes the opening of the sex chromatin during the zygonema-to-pachynema transition at the DSB regions. INO80 also facilitates the recruitment of DNA repair factors at the DSB sites and interacts with DSBR factors to facilitate the silencing of sex chromosomes.

## Results

### INO80 localizes to the sex chromosomes in meiotic germ cells

Immunofluorescence localization of INO80 in pachytene spermatocytes revealed staining on the sex chromosomes (Fig 1A and 1B). Chromatin immunoprecipitation followed by high throughput sequencing (ChIP-seq) for INO80 (GEO Dataset GSE179584) [31] also showed binding on the sex chromosomes (Fig 1C). INO80 binding occurred at the zygotene and pachytene stages (GEO Dataset GSE190590) [32] on both autosomes and sex chromosomes. INO80 localization on the sex chromosomes is more enriched at the postnatal day 18 (P18) pachytene stage (Fig 1C). Additionally, INO80 binding sites on sex chromosomes were present at the DSB sites identified by the meiotic DSB marker γH2A.X (GEO dataset GSE75221) [33]

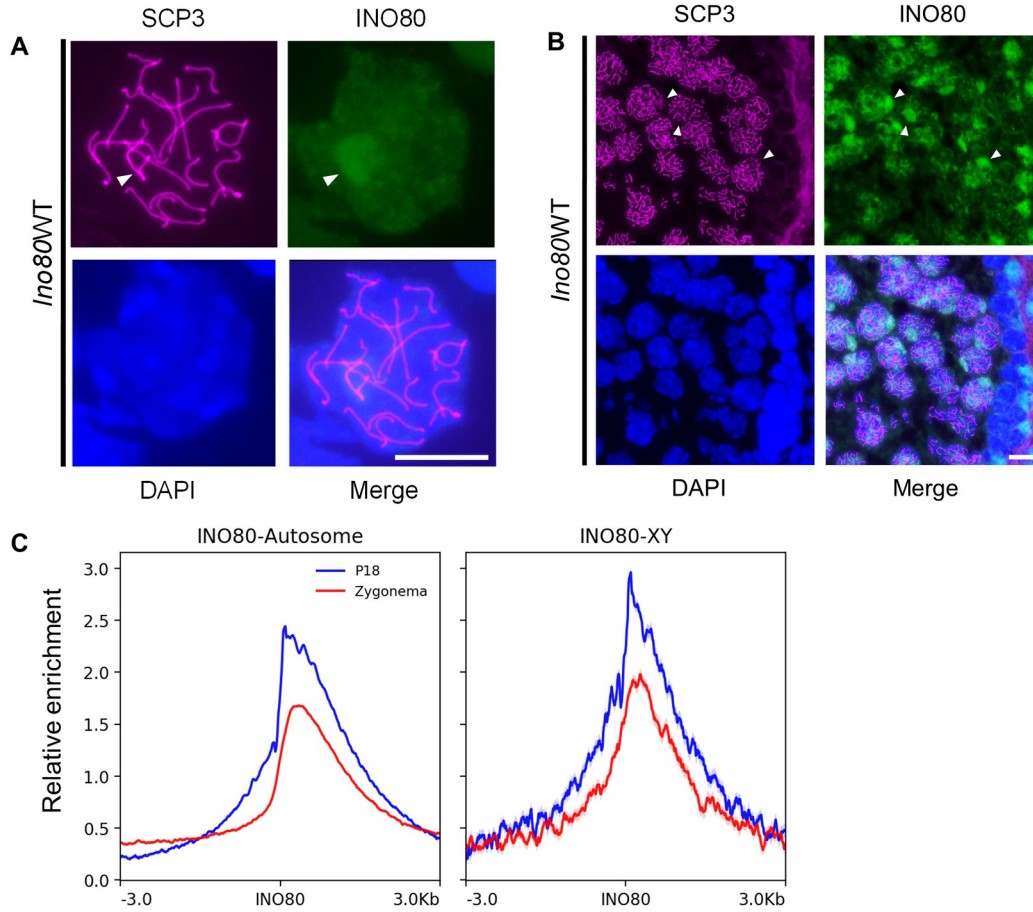

**Fig 1. INO80 localization in the germ cells.** (A,B) INO80 immunolocalization in the pachytene spermatocyte spreads (A) and the seminiferous tubule of the testis (B). SCP3 staining indicates the stage of the meiotic spermatocyte. Magenta: SCP3, Green: INO80, Blue: DAPI. Arrowhead: Sex chromosomes. Scale bar = 10μM. C; Enrichment of INO80 obtained by ChIP-seq on the autosomes and XY chromosomes during zygonema and at postnatal day 18 (P18), where most cells are in the pachytene stage. Blue: P18, Red: Zygonema. (Analyzed from GEO Dataset GSE179584) [31].

(S1A Fig). These binding sites include promoters, intergenic regions, and gene bodies (S1A Fig). Moreover, INO80 and γH2A.X binding in the sex chromosomes were correlated (S1B Fig), where INO80 binding is enriched at the γH2A.X binding sites (S1C Fig). These data suggested a possible role of INO80 in DSB repair and silencing the expression of genes on the sex chromosomes.

## INO80 regulates sex chromosome silencing in pachytene spermatocytes

RNAseq analysis of changes in P18 transcription from autosomes in wild type (*Ino80*WT) and germ cell-specific *Ino80*-null (*Ino80*cKO) spermatocytes (GEO Dataset GSE179584) [31] revealed up-and down-regulated genes (padj< 0.05). However, the mean changes in expression remained close to zero (Fig 2A). In contrast, the differentially expressed genes (DEGs) (padj< 0.05) from the X- and Y- chromosomes remained upregulated in *Ino80*cKO cells (Fig 2A). When plotted along the length of three representative autosomes and the sex chromosomes, like autosomal genes, DEGs from sex chromosomes occur along the length of the chromosomes (Fig 2B). These data confirm that the lack of silencing of sex-linked genes is neither limited to a specific region such as the pseudo-autosomal region and not due to any positional effect. Quantitative RT-PCR analysis validated the upregulation of five representative sex-linked genes (*Ccnb3*, *Nxt2*, *Eda2r*, *Abcd1*, *Usp11*) in *Ino80*cKO spermatocytes (Fig 2C), corroborating the RNAseq data. To investigate further the sex-linked gene expression in pachytene spermatocytes alone, we utilized meiotic germ cell synchronization to isolate homogeneous pachytene spermatocyte population [34] from wild-type and mutant testes and quantified sex-linked gene expression by qRT-PCR. We observed a similar pachytene spermatocyte population in both wild-type and mutant testes (S2A Fig). Synchronized *Ino80*cKO pachytene spermatocytes also exhibited undetectable INO80 levels (S2B Fig). An upregulation of all five genes (*Ccnb3*, *Nxt2*, *Eda2r*, *Abcd1*, *Usp11*) in the synchronized *Ino80*cKO pachytene spermatocytes confirms the lack of suppression of X-linked gene expression during pachynema upon *Ino80* deletion (S2C Fig). Consistent with transcriptional silencing, immunofluorescence analysis of the active form of RNA polymerase II (pSer2) revealed its absence at the sex chromosomes in *Ino80*WT pachytene spermatocytes (Fig 2D–2E). These data correlate with normal transcriptional silencing. In contrast, continued RNA polymerase II (pSer2) immunofluorescence in *Ino80*cKO pachytene spermatocytes indicates incomplete silencing of the sex-linked genes (Fig 2F–2G). Overall, there was a significant increase in RNA polymerase II (pSer2) at the sex chromosomes in pachytene spermatocytes in the absence of INO80 (Fig 2H).

## INO80 is necessary for the localization of DSBR factors to sex chromosomes

DSBR factors such as ATR, γH2A.X, and MDC1 are required to initiate meiotic sex chromosome inactivation in pachytene spermatocytes [6,10,35]. We explored the localization of these DSBR proteins in P21 wild-type and mutant pachytene spermatocytes. Complete sequestration of ATR occurred on early *Ino80*WT pachytene sex chromosomes (Fig 3A and 3C), where relatively consistent ATR binding occurs along the axis at the unsynapsed part of sex chromosomes (Fig 3B and 3D). In contrast, ATR lacked uniform localization on the unsynapsed part of sex chromosomes of early to mid-pachytene *Ino80*cKO spermatocytes (Fig 3E and 3G). Instead, upon quantitation, patchy ATR localization was observed along the axis (Fig 3F and 3H). Further, the overall intensity of ATR at the sex chromosomes was moderately reduced (p<0.05) (Fig 3J). Next, we tested the activity of ATR by immunostaining for the ATR substrate phospho-CHK1 (S345) [36]. Although the localization of pCHK1 (S345) was limited to the sex chromosomes in *Ino80*WT pachytene spermatocytes (S3A Fig), aberrant localization

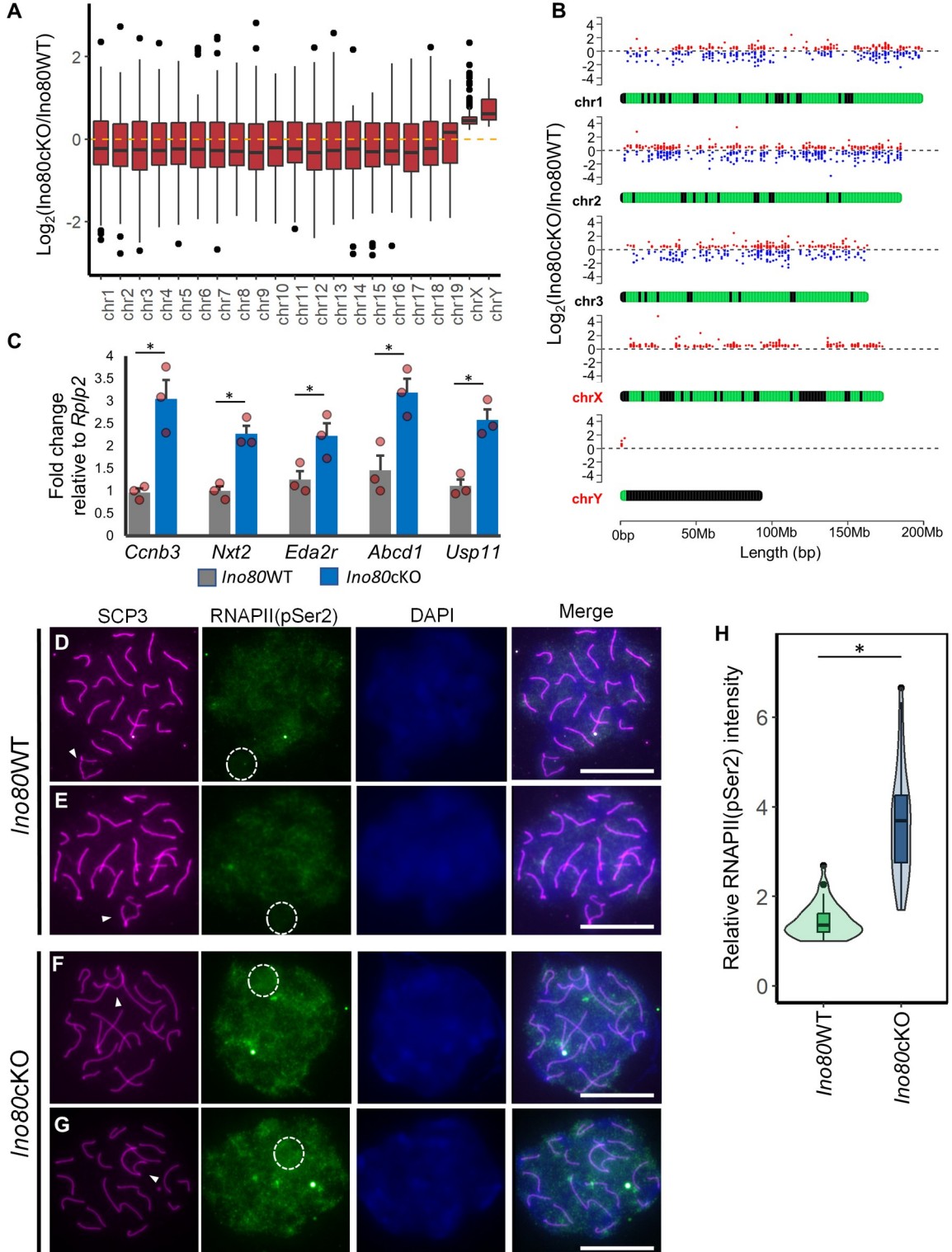

**Fig 2. INO80 required for transcriptional silencing of sex chromosomes during meiotic prophase.** (A) Boxplots showing mean differential gene expression from each chromosome in response to *Ino80* deletion in spermatocytes. The yellow dotted line indicates mean Log2FoldChange = 0. Chr: Chromosome. (n = 5) (Analyzed from GEO Dataset GSE179584) [31] (B) Location of the differentially regulated genes along the length of three representative autosomes and the sex chromosomes. Dots above each chromosome indicate the fold change of individual genes. Upregulated and downregulated genes are indicated by red and blue, respectively. (Analyzed from GEO Dataset GSE179584) [31] (n = 5). (C) Quantitative RT-PCR analysis of representative sex-linked gene expression levels

normalized to *Rplp2* in either *Ino80*WT or *Ino80*cKO testes. Bars represent mean ± s.e.m. *; p<0.05, as calculated by unpaired t-test (n = 3). (D-G) Immunolocalization of SCP3 (magenta) and RNA Polymerase II (pSer2) (green) in spermatocytes either from *Ino80*WT (D,E) or *Ino80*cKO (F,G) (Scale bar = 10μM) showing aberrant localization RNA Polymerase II (pSer2) (RNAPII) in *Ino80*cKO spermatocytes that are indicative of incomplete sex-chromosome silencing. White arrowhead; sex chromosome. The white circle denotes an approximate area of the sex chromosomes. (H) Relative quantification of the immunofluorescence signal intensity of RNAPII quantified from *Ino80*WT (n = 50) and *Ino80*cKO (n = 50) spermatocytes from three biological replicates. *; p<0.05, as calculated by Wilcoxon rank sum test.

occurred in *Ino80*cKO spermatocytes (S3B–S3D Fig). These data indicate aberrant but active DNA binding and kinase activity of ATR in the *Ino80*cKO spermatocytes.

During pachynema progression, the amplification of ATR-mediated phosphorylation of H2A.X depends on MDC1 recruitment [6]. Next, we determined MDC1 localization in *Ino80*cKO pachytene spermatocytes. MDC1 immunofluorescence was visible on early- to late-pachynema *Ino80*WT sex chromosomes (Fig 4A–4C and 4G). Conversely, significantly reduced staining (p<0.05) occurred at the same stages in *Ino80*cKO spermatocytes (Fig 4D–4F and 4G). These results indicate that INO80 facilitates the localization of MDC1 at sex chromosome DSBs during pachynema [23].

To determine whether INO80 physically recruits DSB repair factors, we performed co-immunoprecipitation for INO80 from P21 spermatocytes. The presence of ATR and MDC1 with the INO80-immunoprecipitated samples (Fig 4H) and the detection of INO80 in MDC1- and ATR-immunoprecipitated samples (S4A Fig) supports the physical interaction between INO80 and DSBR factors ATR and MDC1. Adding either ethidium bromide or DNase to the spermatocyte lysate did not alter MDC1 or ATR detection in the immunoblot (Figs 4H and

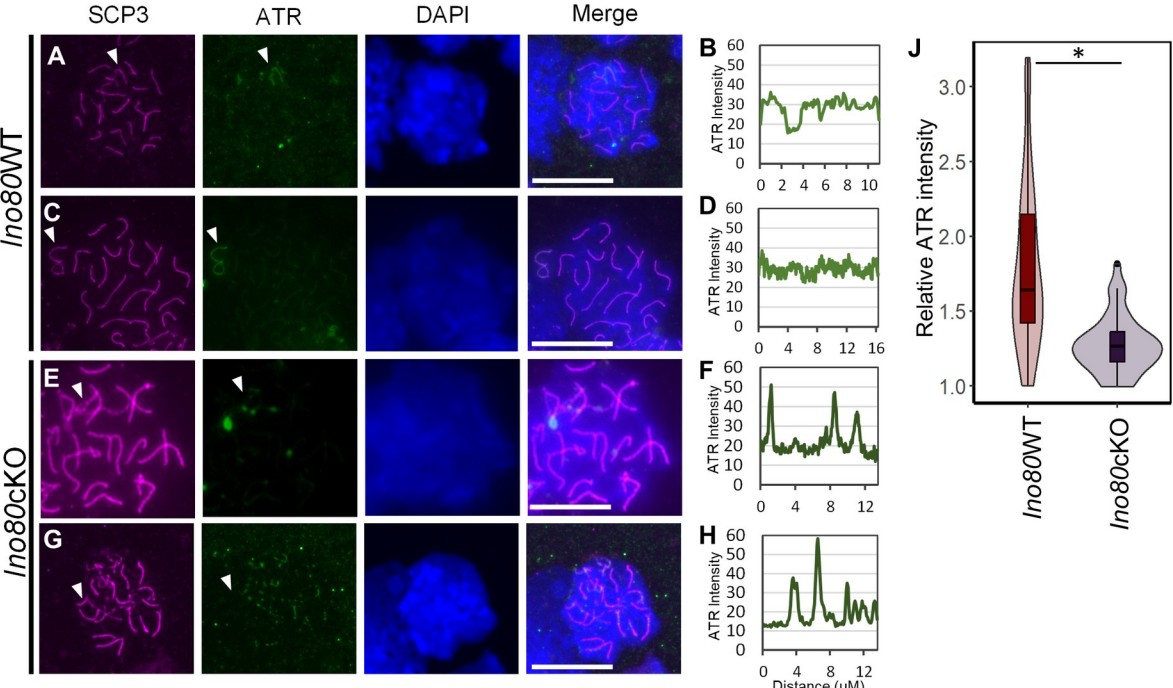

**Fig 3. Aberrant ATR recruitment at sex chromosomes without INO80.** (A-H) Immunolocalization of SCP3 (magenta) and ATR (green) in spermatocytes from *Ino80*WT (A,C) or *Ino80*cKO (E,G). DAPI is shown in blue. Scale bar = 10μM. White arrowhead; sex-chromosome. Line tracing for the quantification of ATR signal along the Y and X chromosome axes in the respective left panels from *Ino80*WT (B,D) and *Ino80*cKO (F,H) are displayed. (J) Relative fluorescent intensity measurement of ATR signal at the sex chromosomes from three biological replicates: *Ino80*WT (n = 53) or *Ino80*cKO (n = 42) pachytene spermatocytes. *; p<0.05, as calculated by Wilcoxon rank sum test.

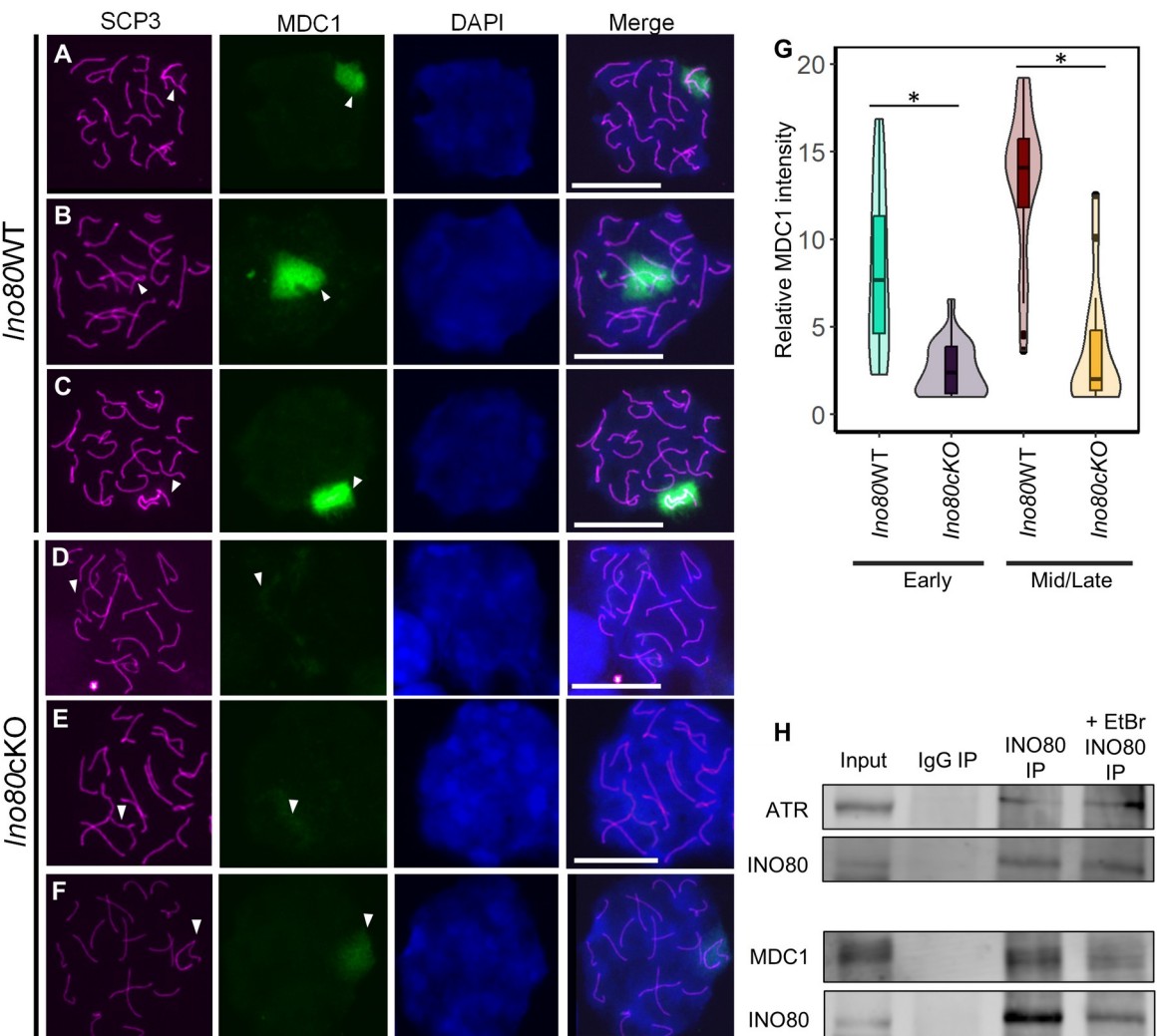

**Fig 4. MDC1 recruitment at the sex chromosomes is facilitated by INO80.** (A-F) Immunolocalization of SCP3 (magenta) and MDC1 (green) in spermatocytes from Ino80WT (A-C) or Ino80cKO (D-F). DAPI is shown in blue. Scale bar = 10µM. White arrowhead; sex-chromosome. (G) Relative fluorescent intensity measurement of MDC1 signal at the sex chromosomes from either *Ino80*WT (early; n = 30, mid/late; n = 39) or *Ino80*cKO (early; n = 23, mid/late; n = 17) pachytene spermatocytes from three biological replicates. *; p<0.05, as calculated by Wilcoxon rank sum test. (H) Immunoblot images demonstrating the interaction of INO80 with ATR and MDC1 by the presence of MDC1 and ATR in INO80 immunoprecipitated *Ino80*WT spermatocyte homogenates and INO80 when immunoprecipitated with MDC1 from *Ino80*WT spermatocyte homogenate.

S4B). This result indicates the interaction between MDC1 and INO80 is chromatin independent.

Further, to determine MDC1 recruitment, we performed cleavage under targets and release using nuclease (CUT&RUN) in synchronized *Ino80*WT and *Ino80*cKO pachytene spermatocytes. We observed robust MDC1 enrichment at INO80-binding sites only in the sex chromosomes in *Ino80*WT spermatocytes. In contrast, reduced MDC1 occupancy was present in Ino80cKO spermatocytes (Figs 5A and S4C). Alternatively, in autosomes, MDC1 enrichment was absent (Fig 5A). MDC1 occupancy was also substantially reduced at and around DSB sites marked by γH2A.X (Fig 5B). Further, genomic annotation analysis of the MDC1 binding sites (S1 Table) suggests majority of MDC1 occupancy was present in distal intergenic regions (63.95%). In contrast, reduced occupancy was distributed among promoters (9.19%), exons

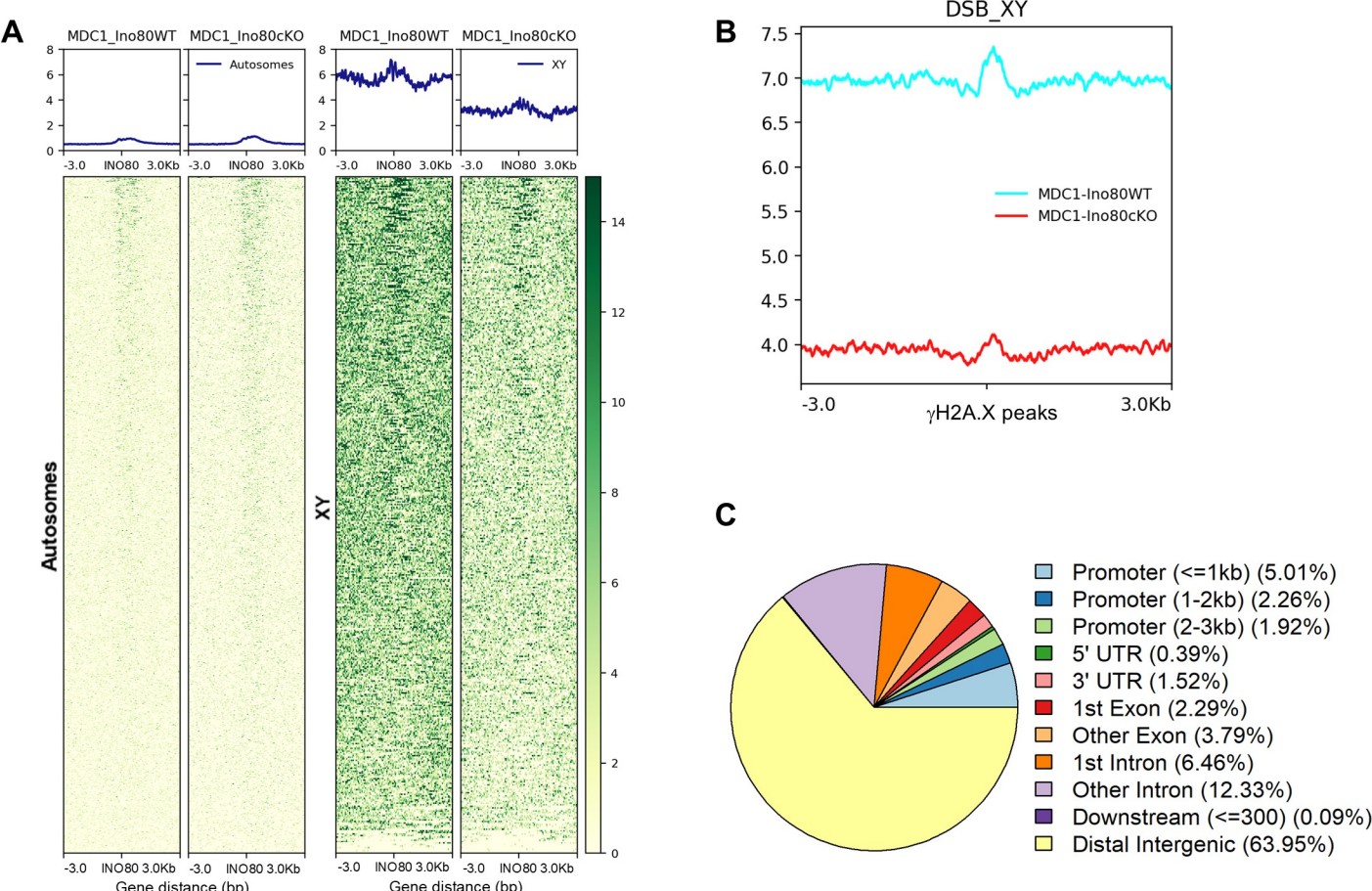

**Fig 5. Genomic occupancy of MDC1.** (A) Heatmap illustrating MDC1 occupancy at the INO80 binding sites in autosomes (left) and sex chromosomes (right). (B) Metaplot illustrating MDC1 occupancy at the DSB sites marked by γH2A.X binding. (C) Genomic annotation of MDC1 peaks in *Ino80*WT spermatocytes.

(6.08%), and introns (18.79%) (Fig 5C). The reduction in MDC1 occupancy in *Ino80*cKO corroborates our previous observation by MDC1 immunofluorescence staining. Further, we also observed the downregulation of MDC1 mRNA and protein in the *Ino80*cKO spermatocytes (S4D and S4E Fig). In addition, the enrichment of INO80 at the promoter of *Mdc1* suggests a requirement for INO80 regulation of MDC1 expression (S4F Fig).

Because MDC1 is essential for amplifying γH2A.X, we determined whether perturbation of γH2A.X distribution occurred in the *Ino80*cKO spermatocytes. γH2A.X staining was observed in the *Ino80*WT pachytene spermatocytes (Fig 6A and 6C), spanning the entire sex body (Fig 6B and 6D). While γH2A.X staining was observed in the *Ino80*cKO spermatocytes (Fig 6E and 6G), its expansion throughout the sex chromosomes was perturbed and more concentrated near the axis (Fig 6F and 6H). In addition, a moderate reduction in the overall γH2A.X signal at the sex chromosomes occurred in *Ino80*cKO spermatocytes, suggesting a perturbed amplification of γH2A.X (Fig 6J).

## INO80 regulates chromatin accessibility at the sex chromosomes

INO80 is capable of histone exchange, most notably removing H2A.Z from chromatin [23,24]. To determine how INO80 changes chromatin dynamics on the sex chromosomes and whether its regulation of sex-linked genes is H2A.Z-dependent, we compared H2A.Z occupancy (GEO

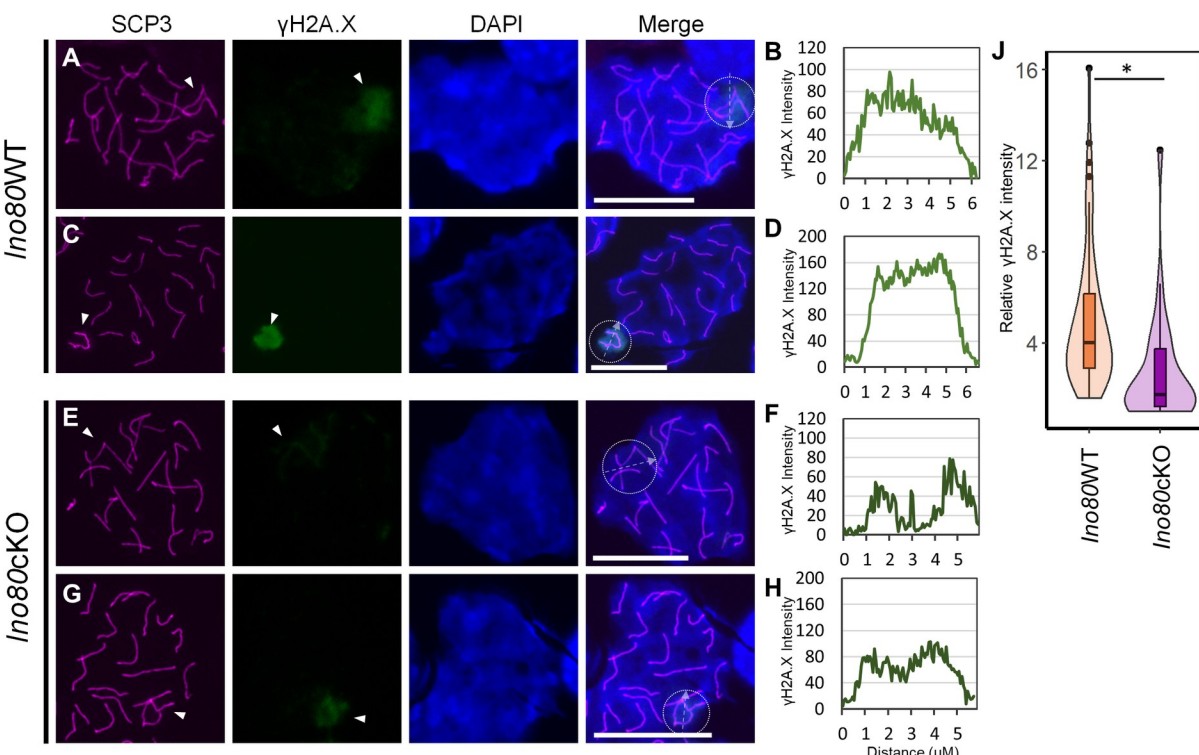

**Fig 6. Aberrant γH2A.X localization in *Ino80*cKO spermatocytes.** (A-H) Immunolocalization of SCP3 (magenta) and γH2A.X (green) in pachytene spermatocytes from *Ino80*WT (A,C) or *Ino80*cKO (E,G) (Scale bar = 10μM). DAPI is shown in blue. Scale bar = 10μM. White arrowhead; sex-chromosome. Line tracing for the quantification of γH2A.X signal along the dotted arrow in the dotted circle (marking the approximate sex body area) in the respective left panels from *Ino80*WT (B,D) and *Ino80*cKO (F,H) are displayed. (J) Relative fluorescent intensity measurement of γH2A.X signal at the sex chromosomes from either *Ino80*WT (n = 58) or *Ino80*cKO (n = 29) pachytene spermatocytes from three biological replicates. *; p<0.05, as calculated by Wilcoxon rank sum test.

Dataset GSE179584) [31] at sex-linked gene promoters in P18 *Ino80*cKO and *Ino80*WT spermatocytes. There was little change in H2A.Z levels at the promoter and transcriptional start sites (TSS) and INO80 binding sites on the sex chromosomes (S5A and S5B Fig). These results suggest that H2A.Z does not mediate the sex-linked gene regulation by INO80. We also examined the occupancy of activating histone modification H3K4me3 and suppressive histone modification H3K27me3 at the sex-linked promoters. Neither showed any change (S5C and S5D Fig) and did not mediate INO80-dependent gene regulation in the sex body.

Chromatin accessibility can be another regulator of DNA-interacting protein binding at the chromatin. A global change in chromatin accessibility occurs in developing spermatocytes during the mitosis-to-meiosis transition and subsequent progression through meiosis [37]. We performed Assay for Transposase-Accessible Chromatin with high-throughput sequencing (ATAC-seq) to determine accessible chromatin distribution in the developing *Ino80*WT spermatocytes at P12 (enriched at zygonema) and compared it with ATAC-seq data from P18 *Ino80*WT spermatocytes (enriched at pachynema) (GEO Dataset GSE179584) [31]. Chromatin accessibility at promoter/TSS regions in autosomes at both P12 and P18 was higher than that observed for the sex chromosomes (S6A Fig). Further, an increase in chromatin accessibility occurred at these regions in both autosomes and sex chromosomes from P12 to P18 (S6A Fig). This increase did not reverse in *Ino80*cKO spermatocytes at P18 (S6A Fig). These data indicate a lack of requirement for INO80 in generating accessible chromatin at promoter/TSS during

this transition. Next, we determined how the distribution of chromatin accessibility changed during the zygonema to pachynema transition. Comparison of sex chromosome-specific ATAC-peaks revealed a small portion of unique sites (1.34%) on P12 spermatocytes. Most P12 peaks are present at P18. The common peaks account for 20% of the P18 ATAC peaks from sex chromosomes, while 80% of ATAC peaks at P18 were *de novo* in nature (S6B Fig). Comparison of ATAC-peaks on sex chromosomes from *Ino80*cKO and *Ino80*WT spermatocytes (GEO Dataset GSE179584) [31] showed only 17% of these *de novo* peaks remain in *Ino80*cKO, and 83% of de novo peaks are lost (S6B Fig). Overall, the increase in chromatin accessibility across sex-chromosomes from P12 to P18 was reduced substantially both at all accessible sites (S6C Fig) as well as INO80 binding sites (S6D Fig). Further, genomic annotation analysis of these ATAC-peaks from sex chromosomes (S2 Table) suggested a significant increase in intronic (9% to 19%) and distal intergenic (24% to 51%) accessible regions during the transition from P12 to P18 (S6E Fig). This result is because 81% of the *de novo* gained peaks occurred in intronic and distal intergenic regions, and only 7% were at promoter/TSS sites (S6E Fig). In *Ino80*cKO spermatocytes, we observed a loss of a similar distribution of peaks. Most of the binding occurred in intronic and intergenic regions, suggesting that INO80 regulates the generation of these peaks (S6E Fig).

Next, we performed a chromosome-specific comparison of accessible regions between P12 and P18 spermatocytes from Ino80WT testes at each peak. These experiments revealed a significant increase in chromatin accessibility at various locations across autosomes and sex chromosomes (Fig 7A). All autosomes showed an increase in chromatin accessibility at most of the differentially accessible (DA) regions (FDR <0.05) (Fig 7A). However, sex chromosomes, especially the X-chromosome, exhibited a greater degree of increase at all the DA regions (FDR < 0.05) (Fig 7A). The genomic locations of the DA sites mainly occur at the intronic and intergenic areas. A minor portion of them occur at promoter-proximal regions (Fig 7B). The distribution of increased and decreased accessible areas (FDR < 0.05) was similar. In contrast, the number of regions with reduced accessibility was minimal. These data indicate that a genome-wide increase in chromatin accessibility occurs during the zygonema to pachynema transition. The increase is more prevalent on the sex chromosomes (Fig 7A).

To determine the role of INO80 in regulating the transition in chromatin accessibility during meiotic progression, we compared the accessible sites in P18 spermatocytes from either *Ino80*WT or *Ino80*cKO spermatocytes (GEO Dataset GSE179584) [31]. We observed a minimal change in accessible chromatin at the autosomes. In contrast, most of the DA regions were located at the sex chromosomes, demonstrating a much larger and significant change in accessibility (Fig 7C). Genomic annotation analyses suggested that these DA chromatin regions mainly belong to intronic and intergenic regions (Fig 7D). To determine how these DA sites were correlated to the change in transcription activity at the sex chromosomes, comparison of the nearest genes from the DA sites to the RNAseq data suggested significant enrichment of these sex-linked genes among the upregulated genes in *Ino80*cKO (FDR <0.05) (Fig 7E). These data indicate that INO80 plays a vital role in regulating the increased chromatin accessibility on the sex chromosome during meiotic progression. We next compared ATAC-signal from *Ino80*WT and *Ino80*cKO spermatocytes at the DSB regions marked by γH2A.X binding (obtained from publicly available dataset GSE75221) [33]. A significant decrease in chromatin accessibility in *Ino80cKO* suggests a less permissive environment for DSBR factor recruitment at the DSB sites (Fig 7F). We have also observed a decreased enrichment of MDC1 at the DA regions in *Ino80*cKO (Fig 7G). These data indicate an essential role for INO80 in recruiting DNA damage repair factors to sex chromosome DSBs by regulating chromatin accessibility.

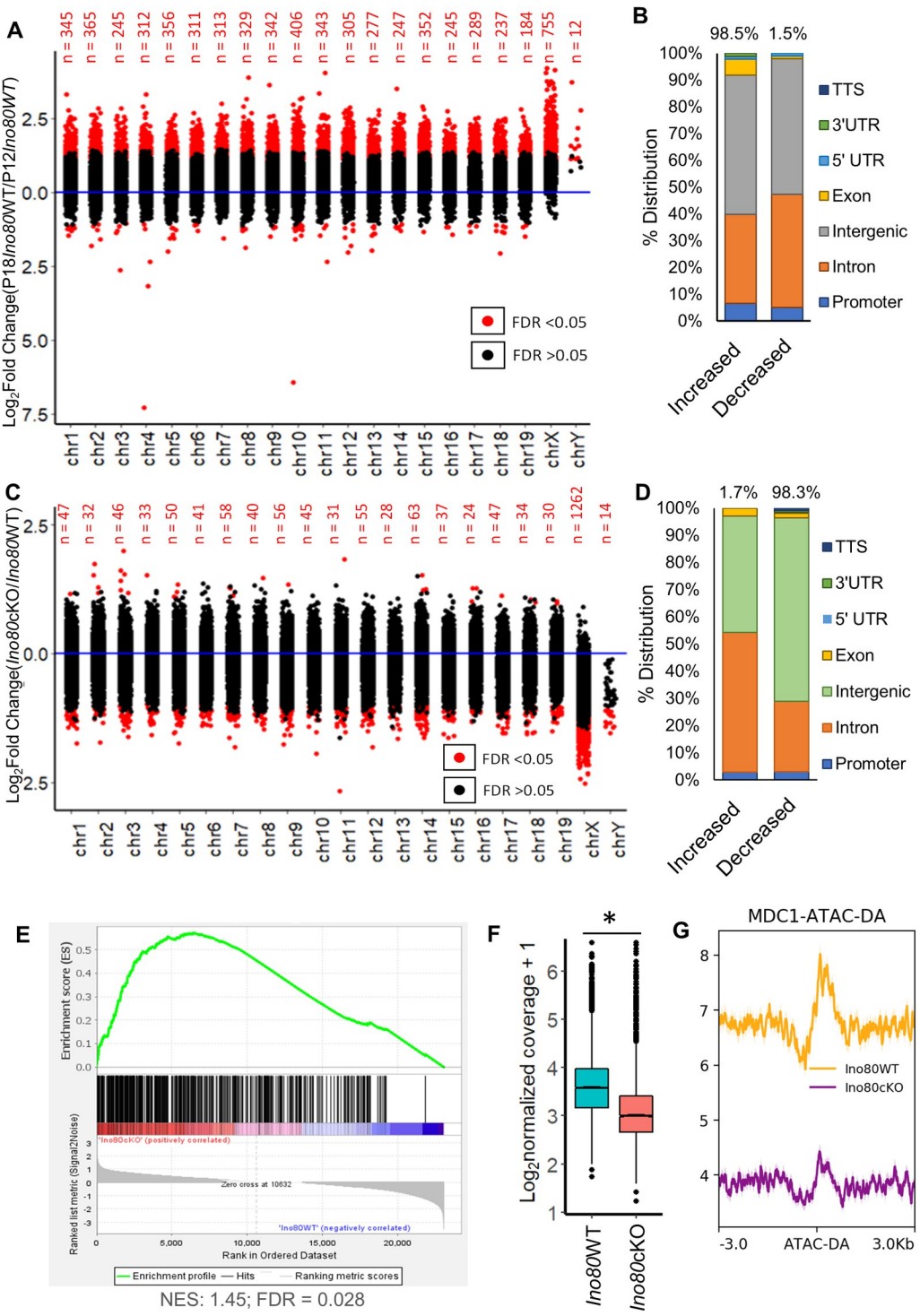

**Fig 7. INO80 regulates chromatin accessibility in spermatocyte sex chromosomes.** (A) Dot plot showing the relative changes in accessibility on each chromosome at P18 pachytene spermatocytes (GSE179584) [31] compared to P12 zygotene spermatocytes. Red dot: FDR < 0.05; Black dot: FDR > 0.05. FDR was derived by Benjamini-Hochberg method (n = 3) (B) Genomic annotation of the differentially accessible regions from (FDR < 0.05) as shown in (A). (C) Dot plot showing relative changes in chromatin accessibility on each accessible site in each chromosome due to *Ino80* deletion at P18. Red dot: FDR < 0.05; Black dot: FDR > 0.05. FDR was derived by Benjamini-Hochberg method (n = 3) (D) Genomic annotation of the differentially accessible regions from (FDR < 0.05) as shown in (C). (D); Genomic annotation of the differentially accessible regions due to *Ino80* deletion. (E) Gene set enrichment analysis with the

nearest sex-linked genes of the differentially accessible regions at sex chromosomes in *Ino80*cKO spermatocytes, depicting enrichment of these genes among the upregulated genes in *Ino80*cKO, determined by RNAseq. NES; Normalized enrichment score. FDR; False discovery rate. (F) Boxplot showing the mean change in ATAC-signal (chromatin accessibility) at all the γH2A.X marked DSB sites [33] between *Ino80*WT and *Ino80*cKO in P18 spermatocytes. *; p<0.05, as calculated by Wilcoxon signed-rank test. (n = 3). (G) Metaplot illustrating the reduction in the MDC1 occupancy at the differentially accessible regions in *Ino80*cKO sex chromosomes.

## Discussion

In this study, we demonstrated a unique role for INO80 in silencing sex-linked genes during pachynema in spermatocytes. INO80 regulates chromatin accessibility at DSB regions of the sex chromosomes. This regulation is independent of the histone variant H2A.Z, a major INO80 effector. We also demonstrated that INO80 interacts with DSBR factors ATR and MDC1, facilitating MDC1 recruitment at the sex chromosomes during pachynema.

An earlier report showed that male germ cell-specific deletion of *Ino80* results in a meiotic arrest phenotype in 8-week-old murine testes. A significant population of pachytene spermatocytes display defects in synapsis and DNA damage response, leading to the loss of spermatocytes in adult testes [18]. However, during the first wave of spermatogenesis in juvenile mice, no significant cell death was observed in *Ino80*cKO testis up to P21. Moreover, unaltered gene expression signatures for zygonema and pachynema stages in *Ino80*cKO spermatocytes at P18, derived from RNAseq data comparing *Ino80*WT and *Ino80*cKo spermatocytes, suggest their relative proportions remain similar, except for a reduced transition from pachynema to diplonema in the absence of INO80 [31]. Utilizing this small window of time, we observed a lack of downregulation of sex-linked differentially expressed genes in *Ino80*cKO spermatocytes, which was also replicated by comparing homogeneous pachynema populations upon synchronization of spermatogenesis. We also validated this overall aberrant sex-linked transcription program in individual *Ino80*cKO pachytene spermatocytes by immunolocalizing active RNA polymerase II at the sex chromosomes. This corroborates the observation of an incomplete MSCI in pachytene *Ino80*cKO spermatocytes. Several other studies also reported similar characteristics that described roles for MSCI-regulating factors [6,38–40].

Several studies have described a regulatory role for INO80 in DNA damage response in somatic cells and meiosis [18,41,42]. The INO80 chromatin remodeling complex interacts with DNA damage factors such as γH2A.X and MEC1 (ATR) at DSB repair sites in yeast [43,44]. We also found that INO80 interacts with DNA repair factors ATR and MDC1 in meiotic spermatocytes. γH2A.X recruitment at DSB sites during zygonema remained intact in *Ino80*cKO spermatocytes [18]. We propose that the initial deposition of γH2A.X at the synapse axis of sex chromosomes initiates INO80-facilitated recruitment of MDC1. This recruitment amplifies γH2A.X in chromatin loops in an INO80-facilitated ATR-dependent fashion.

INO80 is known to regulate transcription in several cell types. It facilitates the recruitment of RNA polymerase II (RNAPII) and its cofactors to the promoters of pluripotency network genes. This recruitment regulates embryonic stem cell pluripotency and reprogramming [26,27]. It can also regulate gene expression by facilitating histone modifications and the exchange of histone variants such as H2A.Z in different cell types [22,23,31,45]. INO80 also regulates somatic gene silencing at the autosomes in spermatocytes by promoting H3K27me3 modification at promoters, while H3K4me3 remains unaffected [31]. However, INO80-dependent silencing of sex-linked genes in meiotic spermatocytes is independent of INO80-mediated direct transcriptional regulation by enabling DNA binding factor recruitment at the promoter-proximal areas.

We showed that INO80 regulates sex chromosome accessibility during meiotic progression. Chromatin accessibility is a central regulator of DNA damage repair response. Less accessible

DNA hinders successful and efficient DNA repair [46,47]. Chromatin accessibility is also essential in transcription factor recruitment and efficient transcription regulation at the promoter-proximal areas [48]. We observed an increase in overall chromatin accessibility at promoter regions of autosomes and sex chromosomes during meiotic progression.

In contrast, a similar level of accessibility remains at the promoter-proximal areas of sex chromosomes due to *Ino80 deletion*. Maezawa *et al.* previously reported that the accessibility at the promoter/TSS during the pachynema transition remains relatively unchanged. At the same time, gene expression changes occur [37]. It is possible that chromatin accessibility is dynamic, and changes occur between stages during spermatogenesis. While the exact mechanism of gene silencing by DSB factors is not clear, the *de novo* generation of accessible regions at the non-promoter areas may provide the open chromatin structure necessary for DSB factors to bind and, therefore, explain the resulting gene silencing by them during MSCI.

Further, recent studies have reported a mechanistic view of the role of INO80 in sliding hexasomes and nucleosomes with different affinities to create accessible DNA [49,50]. It is unclear whether hexasomes facilitate INO80-interaction at the meiotic sex chromosomes. However, it is logical to predict INO80 as a central regulator of sex-linked gene silencing by regulating this *de novo* accessibility generation, as most of these peaks are lost in *Ino80*cKO spermatocytes. The loss of accessibility at the γH2A.X-binding DSB regions and reduced MDC1 occupancy at the DA sites in sex chromosomes due to *Ino80* deletion indicate a possible INO80-dependent recruiting mechanism for DSBR factor MDC1.

Here, we propose that INO80 mediates the *de novo* opening of chromatin around the DSB regions and facilitates recruitment of MDC1, whereby MDC1 initiates γH2A.X amplification by enabling ATR recruitment (Fig 8).

## Materials and methods

### Ethics statement

All animal experiments were performed according to the protocol approved by the University of North Carolina at Chapel Hill's Institutional Animal Care and Use Committee.

### Animals and genotyping

*Ino80* homozygous floxed [18,31] female mice (*Mus musculus*), maintained on an outbred CD1 background, were crossed with *Stra8-Cre*$^{Tg/0}$ males [51] to produce *Ino80*$^{f/+}$; *Stra8Cre*$^{Tg/0}$ males. Male *Ino80*$^{fl/+}$ (*Ino80*WT) and *Ino80*$^{Δ/f}$; *Stra8Cre*$^{Tg/0}$ (*Ino80*cKO) littermates were obtained by crossing *Ino80*$^{f/f}$ females with *Ino80*$^{f/+}$; *Stra8Cre*$^{Tg/0}$ males. S3 Table lists primers used for genotyping. Mice were maintained in an environment with controlled temperature and humidity with 12 h light and 12 h dark cycles and fed *ad libitum*. Spermatocyte synchronization was performed following a published protocol [34]. In brief, newborn pups were treated with retinoic acid synthesis inhibitor WIN 18,446 (Cayman Chemical) (100μg/g), administered orally for 10 days (P1-P10). On P11, one bolus of retinoic acid (100ug per pup) was delivered subcutaneously to initiate spermatogonia differentiation, and pachytene spermatocytes were isolated on P24.

### RNA isolation and quantitative RT-PCR

Total RNA was isolated from Ino80WT and Ino80cKO spermatocytes using Trizol reagent (Invitrogen) followed by the Direct-zol RNA kit (Zymo). Reverse transcription was performed by the ProtoScript II reverse transcriptase (NEB) using random primers. Real-time PCR was performed using Sso Fast EvaGreen supermix (Bio-Rad) on a thermocycler (Bio-Rad). Primers used in this study are listed in S4 Table.

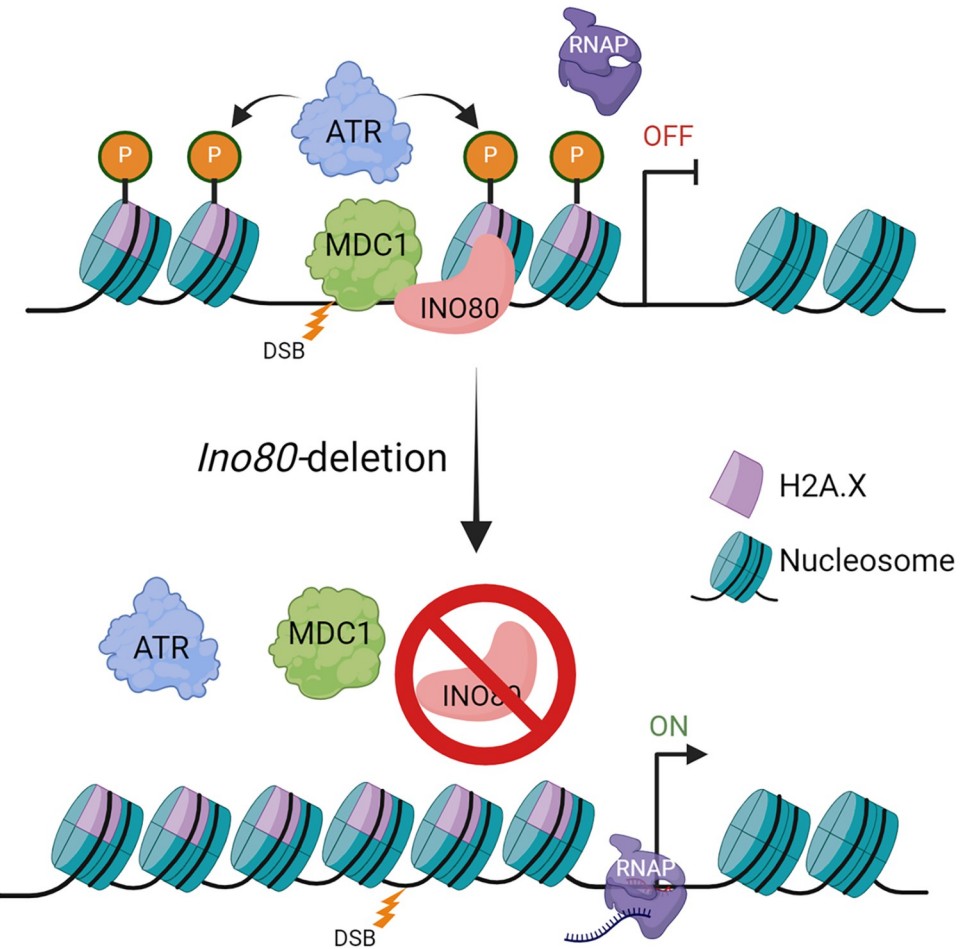

**Fig 8. Schematic illustration of INO80-mediated regulation of meiotic sex-chromosome silencing.** INO80 facilitates MDC1 recruitment at the DSB sites and regulates chromatin accessibility at these regions. MDC1 recruitment allows ATR-mediated amplification of γH2A.X at the chromatin loops. *Ino80*-deletion results in decreased chromatin accessibility and a lack of MDC1 recruitment, which fails to initiate ATR-mediated amplification of γH2A. X.

### Immunofluorescence staining

Spermatocyte nuclear spreads were prepared from freshly harvested testes following a published protocol [52] with modifications [53]. Briefly, single-cell suspensions were prepared from seminiferous tubules. 3ul were added to three volumes of 0.25% NP40 (9ul) on a clean glass slide and incubated for 2 minutes at room temperature. Next, Fixative solution (36ul; 1% paraformaldehyde, 10 mM sodium borate buffer (pH 9.2) was added to the sample, and the slides were incubated in a moist chamber for 2 hours at room temperature. Lastly, slides were dried under a hood, washed with 0.5% Kodak Photo-Flo 200 three times, 1 minute each, and stored at -80°C.

Freshly harvested testes were embedded and frozen in the Optimum Cutting Temperature (OCT) embedding medium to make cryosections. Cryosections (7uM) on glass slides were fixed in freshly made 4% paraformaldehyde solution in PBS for 10 mins at 4°C. Following fixation, samples were washed in PBS 3 times, 5 minutes each, and permeabilized in PBST (PBS + 0.1% Triton-X 100) 3 times for 5 minutes each. Samples were incubated with blocking buffer (10% goat/donkey serum, 2% bovine serum albumin, 0.1% Triton-X 100 in PBS) for 1 hour at

RT before incubation with primary antibody (listed in S5 Table) in blocking buffer overnight at 4°C. The following day, samples were washed 3 times, 5 minutes each with PBST, and incubated with Alexa Fluor-conjugated secondary antibody at room temperature for one hour. Samples were washed once with PBST and counterstained with DAPI, followed by three washes with PBST, 5 minutes each, and mounted in Prolong Gold anti-fade medium (P-36931; Life Technologies). Relative signal intensity was measured by NIH ImageJ software (S6 Table), either at an area marked by a region of interest with assigning the lowest intensity a value of 1 or along a single line transect as described previously [54]. Meiotic substages were identified by either SCP1 and SCP3 immunostaining and/or the shape of the synapsed sex chromosomes [55]. Plots were created in R using ggplot2 [56]

## Isolation of male germ cells

The spermatogenic cells were isolated following a modified version of a previously published protocol [57]. Experiments were performed twice or more with cells isolated from separate mice. Briefly, freshly isolated testes were decapsulated, and seminiferous tubules were digested with collagenase (1mg/ml) in HBSS for 15 minutes at 32°C. Next, the tubules were precipitated by gravity for 5 mins at room temperature and separated, followed by a second digestion with collagenase (1 mg/ml) and trypsin (0.1%) in HBSS for 15 min at 32°C. Trypsin was inactivated by adding equal amounts of soybean trypsin inhibitor. The digested product was pipetted and filtered through 70 uM and then by 40uM cell strainers to get spermatocyte suspension. The spermatocytes were precipitated by centrifugation at 500x$g$ for 10 mins, followed by two washes with HBSS. The cells were finally precipitated and either used in a following experiment or frozen at -80°C for later use.

## Nuclear lysate preparation

Nuclear lysate preparations occurred by isolating spermatogenic cells from P21 testes and incubating the cells in a hypotonic buffer (buffer A:10mM HEPES-KOH pH7.9, 1.5mM MgCl2, 10mM KCl, 0.1% NP-40, 5mM NaF, 1mM Na3VO4, 1mM PMSF, 1x Protease inhibitor cocktail) using 10–20 times the volume of the precipitated cell volume (PCV). After incubation on ice for 15 minutes, the cells were centrifuged at 1000x$g$ for 10 minutes at 4°C. The cells were precipitated and resuspended in buffer A, using twice the PCV, and homogenized with a Dounce 'B' pestle 5 times on ice. Cells were precipitated by centrifugation at 1000 x $g$ at 4°C. Precipitated nuclei were washed in buffer A and resuspended in equal volume lysis buffer (Buffer C) (20mM HEPES-KOH pH7.9, 1.5mM MgCl2, 420mM NaCl, 10mM KCl, 25% glycerol, 0.2mM EDTA, 5mM NaF, 1mM Na3VO4, 1mM PMSF, 1x Protease inhibitor cocktail) for 30 minutes at 4°C on a nutator. The homogenate was cleared by centrifugation at 12000 x $g$ for 10 mins at 4°C, and the supernatant saved in a separate tube. The extraction was performed again from the pellet, and the supernatant mixed with the previous one. The lysate was diluted with 2.8 volume of dilution buffer (Buffer D) (20mM HEPES-KOH pH7.9, 20% glycerol, 0.2mM EDTA, 5mM NaF, 1mM Na3VO4, 1mM PMSF, 1x Protease inhibitor cocktail) to reduce the salt concentration and DTT added as necessary to a final concentration of 1mM. Some samples were treated with either ethidium bromide (50ug/ml) or DNase I (1μg/ml) to inhibit DNA-protein interaction [58] and followed by centrifugation at 12000 x $g$ for 10 minutes at 4°C.

## Co-Immunoprecipitation

Co-immunoprecipitation was performed using 1–1.5 mg proteins from nuclear extracts from the spermatocytes. The lysate diluted with immunoprecipitation (IP) buffer (20mM HEPES-

KOH pH7.9, 0.15mM KCl, 10% glycerol, 0.2mM EDTA, 0.5mM PMSF, 1x Protease inhibitor cocktail) to 1mg/ml concentration. For rabbit antibodies, Protein A conjugated Dynabeads (Invitrogen), and for mouse antibodies, Protein G conjugated Dynabeads (Invitrogen) were used (50ul per sample). Dynabeads were washed in PBS 3 times, 1 minute each, followed by incubation in PBS + 0.5% BSA for 10 minutes. Next, the beads were washed in PBS and IP buffer before use. Samples were precleared with dynabeads for 30 minutes at 4˚C, followed by adding primary antibody (S5 Table) and incubating at 4˚C for 1 hour. Next, dynabeads were added to the samples and incubated overnight at 4˚C. The next day, each sample was washed once in high salt wash buffer (20mM HEPES-KOH pH7.9, 300mM KCl, 10% glycerol, 0.2mM EDTA, 0.1% Tween-20, 1mM PMSF, Protease inhibitor cocktail), twice in IP wash buffer (20mM HEPES-KOH pH7.9, 150mM KCl, 10% glycerol, 0.2mM EDTA, 0.1% Tween-20, 1mM PMSF, Protease inhibitor cocktail), and once in final wash buffer (20mM HEPES-KOH pH7.9, 60mM KCl, 10% glycerol, 1mM PMSF, 1x Protease inhibitor cocktail). Protein elution was performed using 1.3X Laemmli buffer and incubating at 65˚C for 15 minutes, followed by magnetic removal of Dynabeads. The samples were finally heated at 95˚C for 5 minutes to denature and stored at -20˚C until used.

## Western blotting

Protein samples were subjected to polyacrylamide gel electrophoresis followed by overnight wet transfer to polyvinylidene difluoride (PVDF) membranes for fluorescence detection. Blots were blocked by Li-COR intercept blocking buffer followed by primary antibody (S5 Table) incubation overnight in TBS with 0.1% Tween-20. Fluorescent Li-COR secondary antibodies were used for visualization, and blots were scanned using a Li-COR scanner. Uncropped blots are shown in S1 File.

## CUT&RUN

Cleavage under targets and release using nuclease (CUT&RUN) was performed using 250,000 spermatocytes per sample from Ino80WT and Ino80cKO testes following a previously published protocol [59]. Briefly, a single cell suspension of spermatocytes was prepared and immediately washed three times, followed by attachment with concanavalin-A coated beads. These cells were permeabilized using digitonin and incubated overnight at 4˚C with either IgG or antigen-specific primary antibody (S5 Table). The next day, beads were washed twice, followed by protein-A/G-MNase binding in a calcium-free environment. Following two more washes to remove unbound Protein-A/G-MNase, calcium was introduced to start chromatin digestion for 30 mins at 0˚C. Digested chromatin fragments were released for 30 mins at 37˚C. The released chromatin was purified using DNA purification columns (Zymo ChIP DNA Clean & Concentrator). The elute was quantitated, followed by library preparation and high through-put sequencing by Novaseq X Plus.

## ATAC-seq

ATAC-seq was performed following the omni-ATAC method described previously [60], using Ino80WT spermatocytes from P12 testes. 50,000 cells from each sample were washed, and nuclei were isolated. A transposition reaction was performed with these nuclei at 37˚C for 30 minutes in a thermomixer at 1000 r.p.m. followed by a clean-up step using Zymo DNA Clean and Concentrator-5 columns. Libraries were amplified, size selected using 0.5X and 1.8X Kapa pure beads to generate a size range of ~150bp to ~2kb and quantified using NEBNext kit for Illumina. Libraries were pooled and sequenced on a Novaseq 6000 platform, generating 50bp paired-end reads.

## Data analysis

ChIP-sequencing data and CUT&RUN data were trimmed as necessary using trimmomatic [61]. The reads were aligned to mouse reference genome mm10 using Bowtie2 [62] with sensitive settings. Samtools [63] was used to de-duplicate and merge the alignments. Deeptools [64] was used to make depth-normalized coverage tracks and metaplots after removing the mm10 blacklisted regions. Correlation analysis between ChIP-seq datasets were also performed using deeptools multiBamSummary tool using mapping quality >30. MACS2 [65] was used to call peaks using the options -extsize set to 147 and -nomodel.

RNAseq reads were aligned by Tophat2 [66] to mm10, and read counts were obtained by HTseqCount [67]. DESeq2 [68] was used with recommended settings for differential expression analysis. Plots were prepared by ggplot2 [56] and chromoMap [69].

ATAC-seq reads were processed by nf-core/atacseq (ver 1.1.0) pipeline [70]. Briefly, reads were trimmed by Trim Galore (https://www.bioinformatics.babraham.ac.uk/projects/trim_galore/) and aligned to mouse reference genome mm10 using BWA. Picard (https://broadinstitute.github.io/picard/) was used to mark the duplicates, and normalized coverage tracks scaled to 1 million mapped reads were prepared by BEDTools (https://bedtools.readthedocs.io/en/latest/index.html). Differential accessibility analysis was performed by CSAW [71] using region-based binned read count followed by TMM normalization using genome-wide background estimation. Genomic annotation of peaks was performed by Homer [72]. Plots were created in R using ggplot2 [56].

## Statistical analysis

The signals from ATAC-seq, immunofluorescence, and qPCR experiments were compared using the Wilcoxon signed rank test for paired observations, the Wilcoxon rank sum test, or the unpaired t-test for unpaired observations. All the tests performed were two-tailed.

## Supporting information

**S1 Fig. INO80 binding at the DSB sites.** (A) Genomic tracks illustrating INO80 binding at the DSB sites marked by γH2A.X [34] in P18 [32] and zygotene [33] spermatocytes. Each separate genomic location is denoted by alternating background coloring. IG-Up; Intergenic-Upstream. (B) Correlation analysis of INO80 and γH2A.X binding at the X and Y chromosomes. The numbers in the box represent Pearson's correlation coefficient calculated from high confidence reads (mapping quality >30) mapped to either chromosome X or Y. (C) Metaplot showing INO80 occupancy in *Ino80*WT spermatocytes at the sex chromosome DSB sites marked by γH2A.X.
(TIF)

**S2 Fig. Synchronized pachytene spermatocytes exhibit a similar lack of sex-linked gene expression.** (A) Comparison of spermatocyte population in synchronized P24 *Ino80*WT and *Ino80*cKO testes. (B) Immunoblot for INO80 and alpha-actin in synchronized P24 *Ino80*WT and *Ino80*cKO testes. (C) Quantitative RT-PCR analysis of representative sex-linked gene expression levels normalized to *Rplp2* in synchronized P24 *Ino80*WT and *Ino80*cKO testes. Bars represent mean ± s.e.m. *; p<0.05, as calculated by unpaired t-test (n = 3)
(TIF)

**S3 Fig. ATR activity demonstrated by phosphorylation of CHK1 remains intact in *Ino80*cKO spermatocytes.** (A-D) Immunolocalization of SCP3 (magenta) and pCHK1(S345) (green) in *Ino80*WT (A) or *Ino80*cKO (B-D) spermatocytes. Autosomes demonstrate pCHK1 (S345) signal at sites with incomplete synapsis (B), while sex chromosomes at pachynema

exhibit aberrant pCHK1 level in *Ino80*cKO spermatocytes (C-D). DAPI is shown in blue. Scale bar = 10μM. White arrowhead; sex-chromosome.
(TIF)

**S4 Fig. INO80 facilitates MDC1 recruitment and expression.** (A) Immunoblot images demonstrate the interaction between INO80 with ATR and MDC1 by the presence of INO80 in both ATR and MDC1 immunoprecipitated samples. The top panel shows two brightness and contrast levels from the same blot to visualize MDC1 in input and immunoprecipitation samples. Spliced sections in the bottom panel are part of the same blot. (B) Immunoblot images demonstrate the presence of ATR and MDC1 in INO80-immunoprecipitated sample in the presence of DNase I. (C) Genomic tracks depicting the normalized enrichment of MDC1 at the representative sex-linked DEGs in *Ino80*WT and *Ino80*cKO pachytene spermatocytes. (D) Boxplot showing the normalized count of *Mdc1* transcripts from *Ino80*WT and *Ino80*cKO spermatocytes. (n = 5) (Analyzed from GEO Dataset GSE179584) [32] (E) Immunoblot showing MDC1 (top) and α-Actin (bottom) expression from *Ino80*WT and *Ino80*cKO spermatocytes on P18. (F) Genomic tracks illustrated the enrichment of INO80, H3K27me3, and H3K4me3 at the *Mdc1* promoter-proximal area. (Analyzed from GEO Dataset GSE179584) [32]
(TIF)

**S5 Fig. Histone modifications at the sex chromosomes.** Metaplot illustrating changes in H2A.Z (A-B), H3K4me3 (C), and H3K27me3 (D) occupancy at either INO80 binding sites (A) or promoter/TSS regions (B-D) in sex chromosomes (Analyzed from GEO Dataset GSE179584) [32].
(TIF)

**S6 Fig. Chromatin accessibility during meiotic progression.** (A) Metaplot illustrating chromatin accessibility at the promoter/TSS regions of autosomes and sex chromosomes in either P12 *Ino80*WT or P18 *Ino80*WT and *Ino80*cKO spermatocytes. (B) Comparison of ATAC peaks at sex chromosomes during the transition from P12 spermatocyte to P18 spermatocyte and in P18 *Ino80*WT vs. *Ino80*cKO spermatocytes. (C-D) Metaplot illustrating chromatin accessibility at the sex chromosomes in P12 *Ino80*WT and P18 *Ino80*WT and *Ino80*cKO spermatocytes at all the ATAC peaks (C) and at the INO80 peaks (D). (E) Genomic annotation of ATAC-peaks in P12 and P18 spermatocytes, as well as the *de novo* peaks generated during pachynema transition during P18 and lost due to *Ino80* deletion. P18 ATAC-seq data was analyzed from GEO Dataset GSE179584 [32].
(TIF)

**S1 Table. List of MDC1 peaks from MDC1 CUT&RUN sequencing using synchronized spermatocytes at the pachynema stage.**
(XLSX)

**S2 Table. List of ATAC-peaks present in P12 spermatocytes, P18 spermatocytes, peaks that appeared in P18 spermatocytes (*de-novo* peaks) but were absent in P12 spermatocytes, and *de-novo* peaks that were absent in P18 *Ino80*cKO.**
(XLSX)

**S3 Table. Genotyping primers used in this study.**
(DOC)

**S4 Table. Quantitative PCR primers used in this study.**
(DOC)

**S5 Table. Primary antibodies used in this study.**
(DOC)

**S6 Table. List of intensity measurements used in this study from *Ino80*WT and *Ino80*cKO samples immuno-stained with RNAPII, ATR, MDC1 and γH2A.X.**
(XLSX)

**S1 File. Full length images of the Western blots included in the study.**
(PDF)

## Acknowledgments

We thank Magnuson lab members for their input and comments.

## Author Contributions

**Conceptualization:** Prabuddha Chakraborty, Terry Magnuson.

**Data curation:** Prabuddha Chakraborty, Terry Magnuson.

**Formal analysis:** Prabuddha Chakraborty, Terry Magnuson.

**Funding acquisition:** Terry Magnuson.

**Investigation:** Prabuddha Chakraborty.

**Methodology:** Prabuddha Chakraborty.

**Project administration:** Terry Magnuson.

**Resources:** Terry Magnuson.

**Software:** Prabuddha Chakraborty.

**Supervision:** Terry Magnuson.

**Validation:** Prabuddha Chakraborty.

**Visualization:** Prabuddha Chakraborty.

**Writing – original draft:** Prabuddha Chakraborty.

**Writing – review & editing:** Terry Magnuson.

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
