## [Decision Letter · Decision Letter 0]

11 Apr 2024

Dear Dr Magnuson,

Thank you very much for submitting your Research Article entitled 'To the Editorial Staff,

INO80 regulates chromatin accessibility to facilitate suppression of sex-linked gene expression during mouse spermatogenesis' to PLOS Genetics.

The manuscript was fully evaluated at the editorial level and by independent peer reviewers. The reviewers appreciated the attention to an important problem, but raised some substantial concerns about the current manuscript. Based on the reviews, we will not be able to accept this version of the manuscript, but we would be willing to review a much-revised version. We cannot, of course, promise publication at that time.

If you decide to revise the manuscript for further consideration at PLOS Genetics, please aim to resubmit within the next 60 days, unless it will take extra time to address the concerns of the reviewers, in which case we would appreciate an expected resubmission date by email to plosgenetics@plos.org.

We are sorry that we cannot be more positive about your manuscript at this stage. Please do not hesitate to contact us if you have any concerns or questions.

Yours sincerely,

Charles G. Danko, Ph.D.

Guest Editor

PLOS Genetics

Wendy Bickmore

Section Editor

PLOS Genetics

Reviewer's Responses to Questions

**Comments to the Authors:**

Reviewer #1: In their previous paper (Development.2023 150, dev202158), the authors discovered that the absence of INO80 leads to a blockage of spermatocytes at the meiotic stage, along with the localization of INO80 in the XY body. In this paper, they further find that the lack of INO80 results in the failure of MSCI and suggest a possible interaction between INO80 and both ATR and MDC1, further implicating INO80 in the MSCI process. This provides an intriguing perspective on understanding the initiation of MSCI. Notably, INO80 is not a protein that is specifically expressed during meiosis, and I am curious as to why it plays a specific role at this developmental stage, a question that seems unanswered in this paper. Additionally, spermatocytes lacking INO80 appear to exhibit conjoint abnormalities, with many analyzed cells possibly not in the pachytene but in an abnormal leptotene stage. This raises concerns that some conclusions about MSCI anomalies may be due to incomplete development rather than specific involvement of INO80. Therefore, this paper may need to provide more compelling evidence.

Some specific concerns:

1. In Figures 1 A and B, the fluorescent localization of INO80 seems to have been presented in the paper Development (2023) 150, dev202158 and does not constitute new evidence. What does the Y-axis represent in Figure C? The conditions of other chromosomes could also be presented to more clearly delineate the distribution of INO80 across different chromosomes.

2. Although ATR is shown to decrease in the marked area in Figure 3 G, it is noteworthy that other synaptonemal complex regions appear to be upregulated. Moreover, judging from SYCP3, it seems this cell has additional synapsis issues. Please consider whether including such cells in the analysis of pachytene spermatocytes could impact the results. A similar situation exists in Fig.S3.

3. For Figures 4 and S4, it would be better to use cKO as a control instead of IgG.

4. Figure 7 could be revised to reflect the structural model of the Hexasome-INO80 complex revealed in Science 381, 313–319 (2023), to update the depiction of the INO80 action model.

Reviewer #2: In this manuscript, Chakraborty et al studied the role of the INO80 chromatin remodeling complex in sex-linked gene expression during mouse spermatogenesis. They showed that INO80 localize at the XY body in pachytene spermatocytes by immunofluorescence staining. Ino80 deletion leads to the de-repression of sex-linked genes, accompanied by the enrichment of active RNA polymerase II at the sex body and the reduction of DNA double-strand break response factors. Interestingly, Ino80 deletion does not significantly affect the histone variant H2AZ occupancy on the sex chromosomes. Instead, it reduces chromatin accessibility based on ATAC-seq. Together, the authors propose that INO80 is required for DNA repair factor localization to facilitate the silencing of sex-linked genes during meiosis.

This study provides new insights to the role of INO80 and chromatin remodeling in gene regulation and sex chromosome silencing during meiosis. However, many observations are based on immunofluorescence staining and do not have the necessary resolution to fully support the conclusions. The manuscript will be much strengthened if the authors can address the following questions and comments:

1) Fig S1: can the authors provide metagene plots to show whether gamma-H2AX co-localizes with INO80, both on autosomes vs. sex chromosomes?

2) Fig 4H: can the authors carry out INO80 IP in the presence of DNase?

3) Fig S4A: can the authors sequence the MDC1 Cut&Run samples to better examine MDC1 occupancy in WT vs. Ino80 KO spermatocytes?

4) Fig 5: similarly, can the authors carry out gammH2AX Cut&Run-seq to examine its occupancy in WT vs. Ino80 KO spermatocytes? Together with the above, these genomic experiments are necessary to strengthen the conclusion that INO80 is required for DSBR factor recruitment.

5) Fig S6A: since most of gained ATAC-peaks from P12 to P18 are not at TSS, can the authors provide metagene plots to show ATAC-seq signal centered on ATAC-seq peak center (instead of TSS) in P12 WT, P18 WT and P18 Ino80 KO? This will show whether INO80 is required for the enhanced chromatin accessibility at these non-TSS sites.

6) Fig 6C: to better connect the changes in ATAC-seq signal with changes in sex chromosome gene expression after Ino80 deletion, can the authors examine whether de-repressed sex chromosome genes in Fig 2A-B are over-represented in the nearest genes with differential ATAC-seq peaks in Fig 6C?

7) Fig 6E: can the authors provide box plot or metagene plot to show changes in ATAC-seq signal at INO80-bound regions between WT vs. Ino80 KO?

Reviewer #3: “INO80 regulates chromatin accessibility to facilitate suppression of sex-linked gene

expression during mouse spermatogenesis”

Review for PLOS Genetics

In this manuscript, Chakraborty and Magnuson explored the role of INO80 in regulating sex-chromosome gene expression during pachynema in male meiocytes. The authors used an impressive mix of bioinformatics, genetics, immunoprecipitations, and immunostaining to show that INO80 locates at the sex chromosomes during pachynema, is responsible for recruitment of DSBR proteins, binds to areas of open chromatin and DSBs, and negatively regulates expression of X and Y linked genes. The experiments are sound, and conclusions and suggestions are supported. Despite this, the introduction and parts of the discussion should be clarified, and a few small experiments should be performed.

Major Comments:

Introduction

• The introduction is hard to follow. It would help if the authors explained how MSCI and the sex body are related. Paragraphs 2 and 3 could be consolidated and untangled to describe the temporal order of events for clarity.

• INO80 is not mentioned until line 82. It would be helpful to describe this factor and explain why it drew your attention in this study.

• It would also be helpful to start a new paragraph at line 66 “MSCI is initiated…” as it is very important throughout the rest of the paper that the audience remembers that DSBs initiate MSCI.

• The last paragraph is repetitive, and confusing. The statement that INO80 “promotes the opening of sex chromatin during the zygonema-to-pachylema transition” (here and in the abstract) is conceptually difficult when the paragraph is about silencing of sex-linked genes. The authors need to find a way to unpack this sentence.

Results - Fig. 1

- A main finding of this manuscript is that INO80 is enriched at the sex chromosomes (and sex body), in mid/late pachynema. The authors show this with ChIP-seq data but INO80 localization in meiotic spreads throughout prophase I (leptonema, zygonema, pachynema) should demonstrate that INO80 is not enriched at the sex chromosome in early prophase, but only in pachynema.

- Co-staining with other markers of the sex body would validate the findings, and could show which proteins of the sex body become enriched in a similar pattern or with similar timing.

- At the end of figure 1 results, line 100, “These data suggested a possible role of INO80 in DSB repair…” but, I believe the authors actually mean “a role in MSCI initiation”. This is an important distinction.

Line 129 - 131: Is there a sex body in the absence of INO80?

Figure 2 – this figure needs an IF marker for the sex body to show that POL II is not excluded. Is the problem that all sex body markers are gone in the absence of INO80? If this is the case, the authors need to explain that POL II is present in the region of the XY chromosome pair. I don’t think you can say it is absent from the “sex body” if you can’t label it.

Figure 3 – In this figure, images in A,C, and G are dim which raises questions about whether the image is good enough to see ATR if it were present. The nuclei in A and G are very small. Is this significant? Or is the mag different here?

Figure 4F – This figure has a similar problem. The SYCP3 label is dim which raises questions about whether the level of MDC1 is actually lower than wild type. Please replace these images with a different sample.

Figure 5 – Most of the text suggests that the authors believe that INO80 amplifies gH2AX in loops, but in some mutant samples, I don’t see any gH2AX. Do the authors believe that DSBs happen in the absence of INO80? Do DSBs occur at consistent sites? Does Spo11 make these breaks? A little more information about this literature would be helpful. I also found it strange that DNA is not required for interaction of INO80 with MDC1 and ATR. I was thinking that INO80 was attracted to DSBs and recruited the other components of the complex. Other readers likely also need further explanation of this section.

Figure 6 -- The hypothesis that INO80 is needed for opening chromatin at DSBs to recruit other proteins involved in MSCI is very interesting. However, since it is counterintuitive that it is necessary to open chromatin to silence genes, this needs more careful explanation.

Minor Comments:

Line 477: manuscript should be lower case.

Figure 2D: There is a circle in the first panel (SCP3 staining) that shouldn’t be there.

**Have all data underlying the figures and results presented in the manuscript been provided?**

Reviewer #1: None

Reviewer #2: Yes

Reviewer #3: Yes

PLOS authors have the option to publish the peer review history of their article (what does this mean?). If published, this will include your full peer review and any attached files.

Reviewer #1: No

Reviewer #2: No

Reviewer #3: No

---

## [Decision Letter · Decision Letter 1]

3 Sep 2024

Dear Dr Magnuson,

Thank you very much for submitting your Research Article entitled 'INO80 regulates chromatin accessibility to facilitate suppression of sex-linked gene expression during mouse spermatogenesis' to PLOS Genetics.

The manuscript was fully evaluated at the editorial level and by independent peer reviewers. The reviewers appreciated the attention to an important topic but identified some concerns that we ask you address in a revised manuscript.

We therefore ask you to modify the manuscript according to the review recommendations. Your revisions should address the specific points made by each reviewer.

To resubmit, log into your Editorial Manager account and select the option 'Revise Submission' in the 'Submissions Needing Revision' folder.

Yours sincerely,

Charles G. Danko, Ph.D.

Guest Editor

PLOS Genetics

Wendy Bickmore

Section Editor

PLOS Genetics

Reviewer's Responses to Questions

**Comments to the Authors:**

Reviewer #1: The authors have made some revisions based on the reviewer's suggestions, but the following issues still persist:

1. Regarding the stage at which INO80 cKO spermatocytes are arrested, I find the authors' insistence on labeling these cells as pachytene stage to be highly debatable. Despite not providing a detailed classification, the images in Fig. S3 show that INO80 cKO spermatocytes exhibit incomplete synapsis in the SYCP3 signal, which is characteristic of zygotene arrest rather than pachytene. This incomplete assembly of the central element of the synaptonemal complex undermines the foundation of many of the authors' conclusive statements.

2. In Fig. 1C, the comparison of enrichment between the sex chromosomes and autosomes is not compelling. The sex chromosomes do not appear significantly enriched compared to autosomes overall, and some individual autosomes may even show higher enrichment than the sex chromosomes. However, the authors did not provide specific results for this.

3. The role of INO80 in recruiting MDC1 and ATR lacks strong support from high-quality IP data. Additionally, there is an apparent absence of robust co-localization data.

Reviewer #2: The authors have carried out additional experiments and analysis to address questions and comments from the previous review.

Reviewer #3: This is an interesting story and the authors have done a good job of addressing my concerns in the body of the manuscript.

However, the presentation of figures could be improved. The in-figure labeling is minimal and often it is very difficult if not impossible to read.

**Have all data underlying the figures and results presented in the manuscript been provided?**

Reviewer #1: Yes

Reviewer #2: Yes

Reviewer #3: Yes

PLOS authors have the option to publish the peer review history of their article (what does this mean?). If published, this will include your full peer review and any attached files.

Reviewer #1: No

Reviewer #2: No

Reviewer #3: No

---

## [Editor Report · Decision Letter 2]

17 Sep 2024

Dear Dr Magnuson,

We are pleased to inform you that your manuscript entitled "INO80 regulates chromatin accessibility to facilitate suppression of sex-linked gene expression during mouse spermatogenesis" has been editorially accepted for publication in PLOS Genetics. Congratulations!

Yours sincerely,

Charles G. Danko, Ph.D.

Guest Editor

PLOS Genetics

Wendy Bickmore

Section Editor

PLOS Genetics

Comments from the reviewers (if applicable):

**Data Deposition**

http://datadryad.org/submit?journalID=pgenetics&manu=PGENETICS-D-24-00249R2

**Press Queries**

---

## [Editor Report · Acceptance letter]

9 Oct 2024

PGENETICS-D-24-00249R2 

INO80 regulates chromatin accessibility to facilitate suppression of sex-linked gene expression during mouse spermatogenesis 

Dear Dr Magnuson, 

We are pleased to inform you that your manuscript entitled "INO80 regulates chromatin accessibility to facilitate suppression of sex-linked gene expression during mouse spermatogenesis" has been formally accepted for publication in PLOS Genetics! Your manuscript is now with our production department and you will be notified of the publication date in due course.

With kind regards,

Anita Estes

PLOS Genetics

On behalf of:
